# Impact of Cultured Neuron Models on α-Herpesvirus Latency Research

**DOI:** 10.3390/v14061209

**Published:** 2022-06-02

**Authors:** Angus C. Wilson

**Affiliations:** Department of Microbiology, School of Medicine, New York University, New York, NY 10012, USA; angus.wilson@nyulangone.org; Tel.: +1-212-263-0206

**Keywords:** latency, reactivation, HSV-1, HSV-2, VZV, neurons, cell culture

## Abstract

A signature trait of neurotropic α-herpesviruses (α-HV) is their ability to establish stable non-productive infections of peripheral neurons termed latency. This specialized gene expression program is the foundation of an evolutionarily successful strategy to ensure lifelong persistence in the host. Various physiological stresses can induce reactivation in a subset of latently-infected neurons allowing a new cycle of viral productive cycle gene expression and synthesis of infectious virus. Recurring reactivation events ensure transmission of the virus to new hosts and contributes to pathogenesis. Efforts to define the molecular basis of α-HV latency and reactivation have been notoriously difficult because the neurons harboring latent virus in humans and in experimentally infected live-animal models, are rare and largely inaccessible to study. Increasingly, researchers are turning to cultured neuron infection models as simpler experimental platforms from which to explore latency and reactivation at the molecular level. In this review, I reflect on the strengths and weaknesses of existing neuronal models and briefly summarize the important mechanistic insights these models have provided. I also discuss areas where prioritization will help to ensure continued progress and integration.

## 1. Prologue

The ‘fireside chats’ hosted by the late Dr. Randall (Randy) J. Cohrs (1952–2021) were a unique element of the annual Colorado Alpha-Herpesvirus Latency Society (CALS) gatherings. With his signature good humor and faux disorganization, Randy would read aloud suggestions for topics of discussion written on scraps of paper by anonymous meeting participants (see Figure 1). Often these talking points touched on fundamental questions already circulating within the α-herpesviruses (α-HV) latency field. How do we operationally define latency? Why is vaccine development against the human simplex viruses proving to be so difficult? How might neurotrophic viruses contribute to Alzheimer’s disease (AD) and other devastating neurodegenerative disorders? What are the strengths and limitations of the current model systems? With an eye to the future, Randy made sure the voices of trainees and newly independent investigators were granted equal attention. From his perspective, reaching definitive and actionable answers during these convivial sessions was less important than simulating an open dialogue between friends and colleagues.

Randy championed the idea that the scientific enterprise thrives when there is a free exchange of ideas, a viewpoint shared by his long-standing colleague Dr. Donald (Don) H. Gilden (1937–2016). Together Randy and Don established CALS as the embodiment of this philosophy and quickly recruited a diverse family of loyal attendees from all over the world. Randy’s sudden passing in July 2021 was a profound shock and he is sorely missed. In tribute to his substantial contributions to both the substance and practice of virology, this article reflects on a theme that was often touched on during the fireside chats, namely the potential of cultured neuron infection models to bring about major advances in our understanding of α HV latency. By continuing these amicable discussions, I hope to inspire the research community to redouble efforts to embrace and improve on the existing models and to unashamedly draw on new strategies and technologies from other areas of biomedical research. As a field, we need to ensure the new knowledge these models and methodologies will undoubtedly provide is quickly integrated into our shared understanding of the replication biology and pathophysiology of these important viruses.

## 2. Part 1: The Ins and Outs of In Vitro Models

The first tissue and cell culture systems (commonly referred to as in vitro, meaning ‘within the glass’) were introduced in the early 1900s and have been used extensively to this day [1]. Techniques to culture neurons were developed almost immediately and have become increasingly sophisticated with respect to the diversity of neuronal subtypes and different culture environments [2]. This review is focused on neurotrophic α-herpesviruses (α-HV), viruses of clinical and veterinary importance that persist for long periods in the peripheral nervous system through a specialized mode of infection known as latency. There are three α-HV that latently infect humans: herpes simplex virus type 1 (HSV-1), herpes simplex virus type 2 (HSV-2), and varicella zoster virus (VZV), and nearly every person on the planet is eventually colonized by at least one of these viruses. Although live-animal infection models have been used extensively to study latency as well as reactivation, which is the transition back into productive replication, the field is seeing a clear shift towards the use of in vitro models. This has led to major advances in our understanding of the intricate relationship between these viruses and the host neuron and are beginning to reveal the molecular details of how neuronal stress responses alter this relationship allowing the virus to reactivate.

## 3. Impact of Cell Culture Models in Modern Biology

Building on discoveries made in yeast, fruit flies, plants and other genetically tractable organisms, the now widespread use of cultured cells as experimental models, has allowed generations of researchers to tease out the molecular details of biological processes in ways that are very challenging or impossible at the organismal level. One has only to look at the extent of our current understanding of gene expression, genome replication, cell division, macromolecular trafficking, and metabolic homeostasis to appreciate the bountiful fruits of this endeavor [3]. As the sophistication of cell culture models increases their potential as experimental platforms has expanded with in vitro models being used to explore higher-level biological processes such as pluripotency, circadian rhythms, aging and senescence, neuronal networking, and of course, host-pathogen interactions [4,5,6,7,8].

It is probably fair to say that a large fraction of what we know about the replication of HSV-1, HSV-2, and VZV has come from studies using infected cultured cells [9]. Often the cells used have been highly-transformed cell lines such as Vero, HEK293 and HeLa cells that have only a passing resemblance to the cells of normal tissues. These are favored because they are highly-permissive to viral replication, easy to grow and increasingly easy to manipulate. This strategy works because researchers can demonstrate that the in vitro model recapitulates the biological processes of interest, without necessarily reproducing every aspect of what might happen in vivo, meaning in the living human or animal surrogate. This tacit acknowledgement has been critical to the advancement of biomedical research, rendering a wide spectrum of biological processes, including disease states, accessible to descriptive and manipulative studies, and in many cases to some level of biochemical reconstitution. In the case of detrimental traits, the in vitro modeling of processes that lead to disease frequently accelerates the identification and refinement of pharmacological interventions. Once molecular connections are established in cell culture, they can be revisited in live-animal models or when possible, in humans. This transfer of findings from in vitro to in vivo is becoming easier with the advent of rapid genome engineering methods and the growing use of integrative approaches such as systems biology and data-driven epidemiology.

## 4. Neuronal Latency Models

In the last 40 years, a variety of in vitro models have been used to study the mechanisms of α-HV latency and reactivation and are eloquently described in other reviews [9,10,11,12]. The different models can be organized into three broad categories depending on the neuronal cells they are built upon. First are the models that use primary neurons harvested from fully or partially dissociated peripheral ganglia isolated from rats, mice, or chickens [13,14,15,16,17]. Second are neuron-like cells differentiated from transformed cell lines including neuroblastomas of rat or human origin [18,19,20,21], and third are human neurons generated by directed differentiation of either induced pluripotent cells (iPSC) or embryonic stem cell (ESC) lines [22,23,24]. Advantages of primary neurons is that these offer a non-transformed genetic background, and have undergone terminal differentiation and some degree of specialization in vivo. The chief disadvantage is they cannot be amplified and must be continuously isolated from freshly dissected ganglia, which is tedious and expensive. Of note, neurons isolated from embryonic or prenatal animals adapt better to being placed into cell culture but as a consequence, exhibit a less mature neuronal phenotype. As discussed below there is evidence this alters their capacity to establish and maintain α-HV latency.

Neurons generated by controlled differentiation in vitro are easier to obtain in bulk but if derived from an immortal cell line are not considered primary neurons, or if generated from dedifferentiated precursors such as primary fibroblasts, require time-consuming differentiation protocols that in some cases involve an intermediary neural stem cell stage. Fortunately, it is often possible to freeze down intermediary stages and later complete the differentiation as needed thereby accelerating the speed at which studies can be conducted. For completeness it is worth mentioning that a few studies have used non-neuronal cells to establish non-productive infections resembling latency, either through the use of culture conditions such as heat stress that are inhibitory to productive replication, or by infecting with mutant viruses lacking immediate-early genes [25,26,27,28,29]. Although quicker and less expensive to generate than *bona fide* neuronal cultures, none of these non-neuronal models have gained traction, most likely due to concerns about the in vivo relevance of any findings.

The majority of published α-HV latency studies using in vitro models are focused on HSV-1, rather than HSV-2 or VZV, which are equally important human disease agents, if not more so. In the case of VZV, this reflects a poorly understood block to viral replication in cells from rats, mice and hamsters although interestingly, this barrier is reduced in cells from guinea pigs [30,31]. The availability of human neurons generated by guided differentiation of iPSC or ESC should go a long way to redressing this imbalance [24,32]. Careful comparisons of HSV-1 and HSV-2 in murine primary neuron cultures has revealed striking differences in the capacity of sensory and sympathetic neurons to support both produce replication and latency by each virus [33]. Comparative studies will eventually map the viral and host determinants of this selectivity, providing valuable insight into virus-host interactions. Although studied extensively, neurotropic α-herpesviruses of veterinarian importance such as pseudorabies virus (PRV) and bovine herpesvirus 1 (BHV-1) have similarly lagged in terms of in vitro latency studies, presumably because the natural hosts can be accessed experimentally, albeit with some important practical caveats [34,35]. The same rationale applies to simian varicella virus (SVV), a relative of VZV for which there is an animal model albeit with significant financial and ethical limitations [36]. We should expect to see more in vitro latency studies with all of these viruses as cultured neuron models are accepted as legitimate experimental platforms and their usage becomes routine in molecular virology laboratories. Direct virus-to-virus comparisons within the same or similar neuronal models will provide useful information about shared requirements for neuronal factors or processes but might also yield some interesting surprises.

Since 2010, substantial advances have been made using HSV-1 latency/reactivation models based on primary neurons isolated from the superior cervical ganglia (SCG) of prenatal rats and neonatal mice; adaptions of models pioneered by Christine Wilcox and Eugene Johnson in the late 1980’s [17,37,38,39]. Primary SCG cultures are attractive because they can be maintained for long periods using nerve growth factor (NGF) and are considered to be more homogeneous than neurons prepared from other ganglia. Similar models have been established using neurons from other rat peripheral ganglia including dorsal root ganglion (DRG) [38], trigeminal ganglion (TG) [40], geniculate ganglion [41], and vestibular ganglion [42]. Primary neuron infection models have also been established using SCG-, DRG- and TG-derived neurons from both pre-natal and post-natal mice [14,15]. These neurons exhibit different growth factor requirements as well as different capacities to support HSV-1 and HSV-2 latency [15,43,44].

Peripheral ganglia are complex tissues and in addition to the neuronal cell bodies, include large numbers of satellite glial cells (SGCs) and infiltrating immune cells. Typically, these non-neuronal cells (accounting for almost 90% of the cell mass of a ganglion) are removed through the combined action of plating and exposure to mitotic poisons such as 5-fluorouracil and aphidicolin that kill proliferating cells and spare the post-mitotic neurons, which in principle should make up the bulk of the surviving cultures. However, a recent single cell RNA sequencing analysis of rat SCG cultures prepared in this manner found a surprising number of cells displaying transcriptomic signatures of fibroblasts and satellite glia were still present following HSV-1 infection [45]. Conceivably, these cells did not divide during the five or so days of culture establishment and thus like the post-mitotic neurons, which comprised 60–70% of the cultures, were not eliminated. Nonetheless, neither the latency-associated transcript (LAT), a signature of HSV-1 latency in human ganglia, nor viral productive cycle transcripts were detected in the non-neuronal cells suggesting that few if any were infected. Thus, although other cells may be present, neurons appear to be the predominant source of viral genomes and viral gene products.

An area of lingering contention is the use of antivirals during the initial exposure of neuron cultures to infectious virus. This is to prevent unwanted productive replication by the input virus. Even if this happens in just few cells, the resulting superinfection can overwhelm the entire culture before latency is established. While antivirals are not required to establish latency *per se*, they greatly increase the efficiency with which latent cultures can prepared and maintained. Infection in the presence of antivirals also allows for higher infectious doses (typically 1–3 plaque forming units per neuron), which may improve quantitative measurements by lowering the background signal due to uninfected cells. Most infection protocols use acyclovir (ACV), an acyclic guanine analog that that blocks viral DNA replication by inhibiting the HSV-1 DNA polymerase and by acting as a chain-terminator when incorporated into nascent genomes [46]. In some models ACV is supplemented with interferon (IFN) to achieve tighter control [23]. Whether exposure to ACV results in more damage to the DNA of the persistent genomes than is observed with in vivo models is unknown [47]. Methods are available to address this important question, although as will be discussed later we currently lack the tools to distinguish genomes that can actually participate in reactivation events from those that cannot. Careful quantitative studies in mouse models and in humans indicate that individual neurons can carry many copies of the HSV-1 or VZV genome [48,49,50], but as yet there is very little information on how many genomes are competent to engage in either partial (abortive) or complete programs of productive cycle gene expression. Elegant studies using isogenic reporter viruses expressing different fluorescent proteins have convincingly shown that during productive infection only a handful of HSV-1 or PRV genomes are transcribed in each infected cell indicating there is some as yet uncharacterized limitation imposed by the host cell [51,52,53]. Studies are needed to determine whether a similar bottleneck applies to genomes undergoing reactivation in neurons.

Although neurons are highly responsive to IFN, they make very little of their own when infected by viruses [54,55]. Instead, the evidence suggests that IFN produced by other cells acts on the infected neurons to control the virus through an unorthodox autophagic response [55]. Potential sources of IFN include infected non-neuronal cells at the site of primary infection [56] as well as infiltrating immune cells within the ganglia [57]. Along these lines, a recent study showed that exposure of murine SCG neurons to type I IFN did not alter the ability of incoming HSV-1 to establish latency but instead limited the potential for reactivation [58]. This was correlated with the entrapment of viral genomes within promyelocytic leukemia nuclear bodies (PML-NBs) that were formed in response to IFN treatment. Rescue can be achieved by depletion of the host PML protein using RNA interference, but this alone was not sufficient to induce reactivation, indicating that other layers of control such as repressive chromatin or transcription factor availability are also important [58].

Developing methods to consistently establish latency-like infections with wild type virus and without recourse to antiviral treatments is the ambition of many laboratories and this may be easier in some neuronal types than others. In the LUHMES model of HSV-1 latency for example, ACV treatment has been shortened to just the first 48 h [18]. It is not clear if modifying protocols to include only brief exposure to an antiviral will be sufficient to eliminate the concerns discussed above. This observation may also be telling us there is a critical window of opportunity at the beginning of the establishment period when the virus can potentially escape intrinsic neuronal control and warrants further study. Neuron age and the accompanying changes in neurotrophic support requirements may also be an important factor in determining whether antivirals are effective. This age-dependence may reflect known changes in epigenetic control mechanisms and differential growth factor receptor expression as neurons commit to a fully differentiated state [59]. There are indications that primary neurons isolated from adult mice are better able to control HSV-1 or HSV-2 than neurons from the equivalent ganglia or pre-natal or neonatal animals and most excitingly, that latent infections capable of reactivation can be established in adult neurons without the use of ACV or IFN [15] A drawback of extending the maturation period is that it adds to the difficulty and expense of generating bulk cultures and further lengthens experimental turnaround times.

Another strategy to establish latency without antivirals is the use of specialized culture devices to infect neurons via the distal axons but not by the cell body or neurites [16,24,60]. This mimics the physical separation of axon terminals located near the sites of primary infection (often in surface tissues such as the skin, wet mucosa, and cornea) from the neuronal cell body located a substantial distance away in the ganglion [61]. While the viral capsid containing the genome is actively transported to the nuclear pores via microtubule networks, is unclear if the viral transcription factor VP16, which is released from the tegument during uncoating, has the capacity to reach the nucleus. As such a plausible hypothesis is that axonal infection allows the heterochromatinization of incoming genomes in the absence of viral tegument factors that would otherwise antagonize this process to sustain productive replication [61,62]. Unfortunately, there are practical limitations to using axonal infections as a means to routinely establish latency without recourse to antivirals. Diffusion barriers made from cloning cylinders and silicone grease have been used to isolate the axons from the cell bodies of dissociated chick ganglia, but in practice it is difficult to achieve and maintain complete fluidic isolation [16]. Preformed microfluidic devices [63] offer a simpler alternative and have been used successfully to establish latent VZV infections of human ESC-derived neurons that could be subsequently reactivated by sequestering NGF with an antibody, but for reasons that remain unclear, not with HDAC or STAT3 inhibitors [24].

A drawback of chamber devices is that only a relatively small numbers of neurons (at most a few hundred) can extend axons through microgrooves separating the two changes and this limited scale is not conducive to many experimental applications. Regardless of these practical constraints, low inoculum axonal infection of rat SCG neuron cultures with PRV resulted in a latency-like infection that could be sustained for more than 20 days without antivirals and could still be reactivated by superinfection with non-replicating UV-inactivated virus [60,64]. Exposure of the cell bodies to viral particles lacking functional genomes (either UV-inactivated or purified light particles) prevented functional PRV introduced via the axons from establishing a latency-like infection [64]. Infection of neonatal rat DRG neuron cultures with HSV-1 recombinants defective for replication and spread in the absence of antivirals found that viral immediate-early (IE) promoter activity was effectively extinguished within 6–8 days but that viral genomes were retained and could reinitiate IE transcription upon withdrawal of NGF or treatment with the deacetylase inhibitor trichostatin A [65]. This suggests a model in which transcriptional silencing is either established quickly yielding a low copy number latency-like infection or silencing takes place more slowly and permits some replication but is ultimately abortive and again resolves as a latent infection. This is supported by recent studies using a new reporter virus (HSV-1 Stayput-GFP) that is fully competent for DNA replication and gene expression but cannot spread from neuron-to-neuron [66]. With this virus, latent infections of murine primary SCG and TG neurons can be readily established and maintained for extended periods in the absence of antiviral treatments. Cultures can then be efficiently reactivated via the two-step mechanism using a novel combination of LY294002, forskolin, and heat shock.

## 5. Part 2: What Have We Learned?

The use of in vitro latency/reactivation models has yielded a wealth of information on the molecular, and to some extent physical, properties of neurons that allow α-HV to establish the persistent infections that we recognize as latency. A picture is emerging in which intracellular pathways that maintain homeostasis within neurons are also required to maintain the viral latency program. This is consistent with the long-standing idea that latency is imposed on the virus by the neuronal environment but can be antagonized by viral immediate-proteins [67]. This places the spotlight on the first few hours if not minutes after a neuron becomes infected—largely uncharted waters from in terms of direct observation. Maintenance of latency is also an active process requiring continuous intracellular signaling. As the mechanisms responsible for the silencing of productive cycle genes come into focus it will become apparent how these vary between neurons. Ultimately, this will help to explain the neuron subtype preferences of each α-HV and also why reactivation is so asynchronous.

## 6. Maintenance of Latency

In vitro latency/reactivation models, especially those based on rodent primary neurons, have provided a wealth of information on the signaling pathways required to sustain α-HV latency. This knowledge has emerged through manipulations that can be difficult to use in live-animal models such as small molecule inhibitors that would be toxic or difficult administer [13,14,68]. Likewise, lentiviruses have proven to be effective genetic tools to deliver short-hairpin RNAs to deplete critical neuronal proteins require for organismal viability or to express proteins of interest without inadvertently inducing reactivation [58,68,69,70,71]. During development, the survival of sympathetic and sensory neurons is dependent on a signaling cascade initiated by the binding of neurotrophic growth factor (NGF) to the high affinity receptor TrkA [72]. Studies by Christine Wilcox and Eugene Johnson in the late 1980s showed that removal of NGF was an effective inducer of HSV-1 reactivation in primary neuron infection models they had developed [17,37,38]. Subsequent studies in similar rat SCG-derived neuron cultures demonstrated that continuous NGF signaling is required to maintain HSV-1 latency and that transient cessation of translation using a 3-h puromycin pulse triggers reactivation [13,70]. As a result, the cellular serine-threonine kinase AKT (also called protein kinase B) has emerged as a critical nexus (see Figure 2) that couples the control of viral latency to important physiological parameters such as nutrient availability, growth factor signaling, and genome integrity [69]. The three isoforms that constitute AKT help to control the activity of mTORC1, a multiprotein complex with kinase activity that regulates cap-dependent mRNA translation. It is not clear is protein synthesis is required to maintain levels of labile repressive factors or to avoid triggering a stress-response that feeds into the pathways discussed in the next section.

**Table 1 viruses-14-01209-t001:** Inducers of α-herpesvirus reactivation in neuronal latency models.

Inducer	Molecular Target(s)	Latency Model	Refs.
Nerve growth factor (NGF) depletion	TrkA receptor tyrosine kinase	rat SCG, hESC neurons	[13,17,24,38]
dexamethasone	Glucocorticoid receptor (GR)	murine SCG	[14]
LY294002, Wortmannin	phosphatidylinositol 3-kinase (PI3-K)	rat SCG, murine SCG, human LUHMES	[13,14,18]
sodium butyrate (NaB), trichostatin A (TSA)	(histone) deacetylases (HDACs)	hESC neurons, murine DRG	[23,43]
AKT inhibitor VIII	allosteric AKT inhibitor	rat SCG	[70]
rapamycin, PP242	mTORC1-selective inhibitor (cap-dependent translation)	rat SCG	[70]
puromycin	ribosome (global translation inhibitor)	rat SCG	[70]
hypoxia	4E-BP hyperphosphorylation by mTORC1 (cap-dependent translation)	rat SCG	[70]
mirin	Mre11 nuclease activity of MRN complex	rat SCG	[45,69]
forskolin	adenylate cyclase	murine SCG, ND-PC12	[68,73]
8-Bromo-cAMP	membrane-permeable cAMP derivative	murine SCG	[68]
shRNA depletion of NGF signaling or DNA damage response factors	e.g., PDK1, raptor, Ku80, TOP2β, Gadd45β, Gadd45γ	rat SCG	[13,45,69,70]
bleomycin	radiomimetic, generates DNA breaks	rat SCG	[69]
etoposide	forms ternary complex with DNA & topoisomerase II (generates dsDNA breaks)	rat SCG	[69]
HSF1A	activates heat shock factor 1 (HSF-1)	rat SCG	[45]
HSV superinfection	transactivates viral promoters	rat SCG, chick eTGE	[16,60]
capsaicin	vanilloid receptor-1 (VR-1), Ca2+ flux	rat DRG	[40]
hexamethylene bisacetamide (HMBA)	broad spectrum kinase inhibitor	chick eTGE	[16]

## 7. Mechanisms of Reactivation

In humans, natural α-HV infections are characterized by periodic reactivation events often linked to environmental and physiological insults that include sunburn, tissue damage, nerve resection, hormonal changes, psychological stress, toxin exposure, and the response to other infectious agents [74]. The identification of specific treatments that induce reactivation in various neuronal culture models has been instructive in terms of exposing the underlying circuitry that unites a seemingly amorphous collection of triggers. The advantage of cell-permeable inducers (see Table 1) is that they can be applied to cultures in a consistent manner, typically by supplementing the culture medium using chemically-synthesized compounds obtained from a commercial source. This standardization should enhance cross-lab reproducibility. Additionally, many inducers act on defined molecular targets that function within well-studied cellular processes such as intracellular signal transduction pathways, homeostatic regulation of protein synthesis and DNA damage response relays.

Interruption of NGF signaling using the PI3 kinase inhibitor LY294002 has been used extensively to reactivate HSV-1 in both rat and murine SCG models [13,14]. This has led to a two-step or bi-phasic model (see Figure 3) to account for the transition from a transcriptionally repressed state to full expression of the productive cycle genes [71]. Viral produce cycle gene expression begins with a synchronous wave of all productive cycle viral transcripts regardless of kinetic class termed Phase I that peaks at 18–20 h post-stimulus. There is evidence that Phase I may not require new viral protein synthesis or transactivation by VP16. Instead, neuronal factors including the stress-response kinases dual leucine-zipper kinase (DLK) and c-Jun N-terminal kinase (JNK) are required [14]. Phosphorylation of serine-10 on histone H (H3S10P) by JNK results in the so-called ‘methyl/phospho switch’, a curious combinatorial mark that allows transcription by RNA polymerase II (RNAPII) in the presence of the otherwise repressive histone H3 lysine-9 trimethyl (H3K9me3) histone modification [75]. Despite detectable late gene expression, there is no evidence that any viral DNA replication occurs in Phase I and viral promoters remain associated with heterochromatin, raising the possibility that genomes that are ‘animated’ in this manner but can easily return to the previous latent state. Lastly, it is apparent from highly sensitive RNA FISH that initiation of phase I does not occur simultaneously in all latently infected neurons [45].

The second wave of viral lytic gene expression, Phase II, occurs approximately 48 h post stimulus. This is characterized by higher levels of viral gene transcription, onset of viral DNA replication and ultimately production of infectious virus. In contrast to Phase I, the repressive heterochromatic marks are removed and it seems likely that active euchromatic marks such as di or tri-methylated histone H3 lysine-4 (H3K4me2/3) are installed, consistent with robust viral gene transcription. As has been extensively described for acute replication in non-neuronal cells, viral protein synthesis, VP16-mediated transactivation, and viral DNA replication are all required for Phase II. Recent single-molecule RNA FISH analyses support the notion that Phase I and Phase II are sequential events and that individual genomes transcribe multiple viral genes during Phase I [45]. The biphasic program is not unique to reactivation in response to LY294002 but can be observed with mirin a small molecule inhibitor of the MRN complex which results in topoisomerase 2-mediated DNA breaks [45,69], as well as in response to increased neuronal excitation [68]. Neuronal hyperexcitability is associated with prolonged stress conditions and IL1-β release, and can be mimicked in vitro using the natural bicyclic compound forskolin, which acts within the neuron to activate adenylate cyclase. The accompanying rise in intracellular cAMP levels also results in DLK activation. Importantly ex vivo (meaning ‘outside of the living body’) reactivation in explanted TGs from latently-infected mice, also begins with a DLK-dependent but histone demethylase-independent wave of viral gene transcription that is essentially indistinguishable from Phase I as defined in SCG neuron cultures [76]. It is notable that in the explant model, viral productive cycle mRNAs can be detected by 5 h rather than 18 h consistent with known differences in the kinetics of DLK activation in response to the extreme trauma associated with axotomy [77].

It is important acknowledge that other reactivation pathways have been proposed based on in vivo models but have yet to tested directly in the context of the in vitro models considered here. The best characterized invokes the selective expression of the viral transcription factor VP16 (encoded by UL48) after exposure to hyperthemic stress in the mouse ocular infection model [78]. VP16 is recruited to enhancer elements associated with the five viral immediate early genes which include ICP0, a chromatin modifier, and ICP4, a sequence-specific transcription factor required for early and late gene expression, and thus should be sufficient to overcome epigenetic silencing. Indeed, expression of VP16 from an adenoviral vector is a potent inducer of reactivation in the rat SCG model without the need for another stimulus [79]. The capacity to reactivate in response to hyperthermal stress maps to the UL48 promoter, which contains sequence motifs recognized by several cellular transcription factors involved in various stress responses [80]. Whether DLK/JNK are required for VP16-induced reactivation is not known. It is possible that stress-kinases are themselves activated by hyperthermic stress and facilitate RNAPII transcription from VP16-responsive viral promoters. Although ectopic VP16 is sufficient to induce reactivation in vitro, the levels of expression are probably much higher than in vivo and might elicit a DLK/JNK stress response. Finally, another potentially distinct in vivo reactivation stimulus is trans-corneal iontophoresis of adrenaline/epinephrine in the rabbit ocular model [81,82]. As with changes in core body temperature it is not entirely clear how exposure to secreted hormones alters intra-neuronal signaling and whether this reactivation pathway also feeds through the DLK and JNK.

The ability to reactivate is an essential property of latency [83], and models based on the neuroblastoma line SH-SY5Y [12] or immortalized HD10.6 cells [20], have encountered difficulties achieving robust reactivation. While this is frustrating given the potential benefits of these more scalable platforms, there might be value in understanding why latent virus is not fully responsive. Does poor reactivation reflect a defect in the neuronal response to the reactivation stimulus itself or some aspect of the chromatin-associated with genomes? Monitoring DLK/JNK activation, Phase I transcription and viral DNA replication in Phase II will help to distinguish defects in signaling from downstream transcriptional events.

## 8. Part 3: The Future

As the use of in vitro α-HV latency models increases it seems a good time for the community as a whole to consider priorities. How many different neuronal models do we actually need? What features should to be developed or refined to best tackle the major gaps in our knowledge? How do we apply the findings from neuronal models to live-animal models and if possible, to humans? The sections below will discuss various opportunities for consolidation or broader development.

## 9. Seeking Consistency

A shared aspiration of research scientists is to achieve consistent results that can be replicated and expanded upon by others [84]. With the proliferation of neuronal infection models there is a risk that poorly understood differences in these models will hamper progress by generating findings that do not carry over from one model to the next. Do contradictory findings reflect weaknesses of the models, differences in experimental design or unrecognized complexity in the biology? How much does it matter if a model does not perfectly mirror everything that happens in experimental animals or humans? Regardless of these uncertainties, the ability to replicate key findings in more than one model system—which can be both time consuming and expensive—needs to be recognized as a strength by reviewers of manuscripts and grants. Too often efforts in this direction are dismissed as showing a ‘lack of innovation’ rather than evidence of experimental rigor.

As a field we need to consider the strengths and weaknesses of consolidation versus diversification. Consolidation means that a small number of neuron sources and/or infection protocols gain widespread usage. A benefit is that favored models become more extensively characterized and the likelihood that new findings are independently validated will increase. The downside is that favored models might unwittingly omit important features of latency in humans or experimentally tractable small-animal models such as mice and rabbits. This may not be so detrimental to mechanistic studies, but could impede progress in the development and testing of therapeutics. Diversification might mean that the number of different models could grow to a point where almost every laboratory uses its own model to the exclusion of others. As a result, generalizable conclusions may be obscured by the idiosyncrasies of individual models. This could impact the consolidation of knowledge and reduce the impetus towards further validation through animal or human studies.

So, what can be done to ensure consistency? This is where a community approach might be fruitful. Investigators could enter into collaborations in which they deliberately try to validate or refute key findings in different models without the need to necessarily break new ground. This of course might draw the criticism mentioned above. Cross-model comparisons could be simplified by the sharing of reference viruses. These might represent the most frequently used laboratory strains or engineered reporter viruses than can be used to establish benchmarks in terms of infectivity, replication efficiency, ease of maintenance in a non-productive infection state and lastly, their ability to reactivate in response to defined stimuli. The importance of strain background is well understood in animal infection models and is likely to hold true in cultured neurons. RNA sequencing of productively infected SH-SY5Y neuroblastoma cells has shown that different HSV-1 strains impact the neuronal transcriptome in different ways and can manifest as different effects on cell morphology, cell-cell interactions and the relative expression of viral proteins [85]. With respect to latency, careful comparisons in the human LUHMES model found a profound difference between HSV-1 strain 17syn+ and KOS(M) in terms of reactivation in response to the phosphatidylinositol 3-kinase (PI3-K) inhibitor LY294002 [86]. Understanding the molecular basis for these differences could be instructive but also potentially distracting in terms of model characterization and data sharing.

## 10. Seeking Scalability

As discussed above, a major challenge to primary neuron models is the practical difficulty and expense of generating infected cultures in sufficient bulk for many biochemical applications. Informative techniques such as PAR-CLIP, ribosome profiling (Ribo-seq), and proximity-labeling all benefit from larger quantities of sample and also require several biological replicates. The same is true for chromatin immunoprecipitation (ChIP), especially when combined with next generation sequencing (ChIP-seq). ChIP-seq has been used to good effect in studies of acute HSV-1 infections [87], non-productive HSV-1 infections in non-neuronal models [88], profiling of insulator factors associated with latent HSV-1 in murine ganglia [89]. The power of this approach is clear from studies of latency and reactivation of lymphotropic γ-herpesviruses, where tens of millions of latently infected cells can be readily obtained [90]. ChIP-seq protocols require the isolation of cross-linked chromatin prior to fragmentation and antibody-mediated capture of the proteins of interest and it is notoriously difficult to achieve consistent chromatin preparations when working with small numbers of cells. Models based on the differentiation of proliferating cells lines such as LUHMES or on proliferating iPSC or ESC precursors offer tremendous promise assuming that sizeable cultures of fully differentiated neurons can be generated on a consistent basis. Alternatively, investigators may opt for targeted nuclease strategies requiring less material and with a better signal-to-noise ratio than standard ChIP-seq [91,92].

## 11. Next Generation Models

It is likely we will see more studies using α-HV latency models based on genetically-isogenic populations of human neurons exhibiting transcriptomes resembling those of *bona fide* sensory or sympathetic neurons. Because of the perceived value in regenerative medicine, directed reprogramming techniques capable of converting untransformed cells such as primary fibroblasts into functional neurons are becoming more efficient and more nuanced in terms of the neuronal subtypes generated, and do not require cells to pass through an embryonic phenotype [93,94,95,96]. However, there are still significant challenges relating to the length of time required to generate usable cultures and unresolved difficulties in achieving desirable uniformity in sufficient quantity. Single-cell RNA-sequencing of more than a hundred iPSC-derived sensory neuron cultures found substantial variability in the gene expression profiles between cultures, more that is detected in postmortem DRG from human donors [97]. Thus, despite the fact that iPSC-derived sensory neurons express classic neuronal markers and exhibit the expected morphology and electrophysiological properties, they are still transcriptionally distinct from the mature neurons encountered by virus during natural infections. The extent to which these differences matter in terms of virus-host interactions needs to be determined but should be kept in mind by those embarking on human in vitro models. Some differentiation protocols generate neurons that more closely resemble the CNS and may serve as excellent models to study the consequences of spread from the PNS into the CNS resulting in acute but very severe outcomes such as herpes simplex encephalitis (HSE) or longer-term pathologies such as AD and AD-related dementias [98].

Recent studies are also beginning to explore the consequences of switching from traditional two-dimensional (2D) cultures into three-dimensional (3D) structures termed organoids [99,100]. Aggregates provide more points of neuron-to-neuron contact in addition to more contacts between neurons and the extracellular matrix. Work in other areas has found that organoid models are better predictors of drug responses in vivo than conventional 2D monolayers. As witnessed by recent efforts to model SARS-CoV-2 infections using liquid-air interface cultures that mimic the epithelial layers of the airway and lungs, organoids are increasingly valued in studies of host-pathogen interactions [101].

Ultimately it might possible to develop mixed cultures composed of neurons and satellite glial cells (SGCs). In peripheral ganglia, the SGCs are found tightly wrapped around the neuronal cell body or soma and are encased within a sheath of connective tissue [102]. The activity of the neurons and SGCs is reciprocally modulated via gap junctions acting in concert with secreted proteins and small-molecules [103]. At a minimum, these interactions could enhance the ability of neurons to enforce α-HV latency by reducing the stresses associated with in vitro culture. Latently-infected ganglia also contain infiltrating T-cells, macrophages and keratinocytes, which again are absent from most cultured neuronal models [104]. These immune cells are known to be important contributors to neuroinflammation and neuropathic pain [105]. There is clear evidence from mice that infiltrating cytolytic T-cells suppress productive replication of HSV-1 through a noncytolytic involving secretion of α-interferon and granzyme B, which degrades ICP4, an essential viral transcription factor [106,107]. As already discussed, it may be difficult to achieve sufficient scale in these complex but perhaps more realistic models.

## 12. Looking to the Horizon

With the growing sophistication of in vitro latency studies, we can expect rapid progress in several key areas. For example, there are still substantial gaps in our knowledge of the chromatin associated with viral genomes, especially during the establishment period. Precision studies are needed to determine if there are additional changes in chromatin composition or post-translational modification over time that can explain why reactivation is less efficient after sustained periods. Studies using the new spread-deficient reporter found that although genome copy number and LAT levels remained relatively constant, the proportion of neurons undergoing reactivation at 30 days post-infection was much reduced compared to 8 or 16 days [66].

With the ability to infect a higher proportion of neuron cultures than occurs by natural infection in vivo, it has become much easier to ask if the transcriptome and proteome of host neurons is changed due to the presence of latent virus and/or the accumulation of latency-associated RNAs. The establishment process has been especially difficult to study in live-animal models because of the spatial and temporal disconnect between primary infections at peripheral sites such as the eye or footpad and the neuronal nuclei situated in the peripheral ganglia. In animals, colonization of TGs is highly asynchronous with new infections and superinfections taking place over several days [80]. As such it is uncertain if incoming genomes undergo limited replication before they are incorporated into heterochromatin and whether the act of replication influences this process in any way.

Although we have a relatively complete picture of the stepwise transition from latency into productive replication, it is still unclear how JNK and other stress kinases are actually targeted to the chromatin surrounding viral promoters. Conceivably this requires neuronal factors, which may themselves be regulated by stress signals. The specific details are important because they might lead to prophylactic strategies to limit reactivation in humans during periods of immunological vulnerability. Another open question is why viral DNA replication is not detected until Phase II, even though many viral mRNA and proteins are synthesized in Phase I. In vitro models will allow systematic analysis of the relative abundance and localization of the largely virus-encoded DNA replication machinery, perhaps revealing unforeseen differences in the control of viral DNA replication in post-mitotic neurons compared to proliferating cell types.

Finally, α-HV latency is characterized by the sustained expression of the latency-associated transcripts (LATs) or the VZV latency-associated transcripts (VLTs) [108,109]. By analogy to the cellular functions of long non-coding RNAs (lncRNAs), these may act *in cis* to influence the chromatin of the viral genomes they are transcribed from and again, in vitro models will be instrumental in identifying any molecular targets [110]. The same is true for host and viral microRNAs which are strongly implicated in dampening productive cycle gene expression [111,112]. Use of human rather than rodent neurons will be beneficial if perfect nucleotide complementarity is critical.

The organization of latent α-HV genomes in terms of the distribution of factors and modifications reflective of facultative heterochromatin and/or euchromatin as well as differences in topology and subnuclear location remain very much a black box. Cryptic heterogeneity may help to explain why only a small subset of latently-infected neurons reactivate in vitro even though the entire culture is exposed to the same stimulus. Does this reflect neuron to neuron differences in the numbers of reactivation-competent genomes or some other variables? Indeed, it is still unclear if multiple genomes within an individual nucleus initiate productive cycle gene transcription or whether a single responsive genome is sufficient.

The availability of sensitive single-cell imaging techniques including the ability to localize and characterize nascent RNA will help to resolve these important questions. Single-cell RNA sequencing (scRNA-seq) has emerged as a highly effective tool to identify and characterize cells by their individual transcriptomic signature and can yield dynamic information through RNA velocity modeling, metabolic labeling and other refinements [113,114]. Use of scRNA-seq to study α-HV latency has been hampered by an inability to reliably detect low abundance viral RNAs or even detect abundant latency products such as the stable LAT introns and mature microRNAs that lack poly(A) tails. Fortunately, sensitivity has greatly improved with the introduction of newer chemistries such as Chromium v3 from 10× Genomics [115]. Using this approach, a recent study was able to detect sufficient numbers of HSV-1 productive cycle transcripts in rat SCG neurons treated with LY294002 to distinguish neurons undergoing viral reactivation from either uninfected neurons or latently-infected neurons that had not responded to the stimulus [45]. This analysis identified a small number of host mRNAs that were upregulated by the stress stimulus but interestingly, only in virus-infected cells. Prominent among these were mRNAs encoding components of the heat shock response pathway and the Gadd45 protein family. Depletion of Gadd45b, and to a lesser extent, Gadd45g, increased the frequency of reactivation. Surprisingly, this was insensitive to JNK inhibition hinting at a mechanism that is less reliant on the methyl/phospho switch. In support of a suppressive role, direct expression of Gadd45b from a lentiviral vector reduced expression of both ICP4 and viral late transcripts in response to loss of NGF signaling (LY294002) or dsDNA break repair (mirin). Taken together these new studies identify Gadd45b as an intrinsic restriction factor that limits reactivation by suppressing viral late gene expression.

Interestingly, single neuron immunofluorescence imaging (shown schematically in Figure 4) revealed that the Gadd45b protein is found throughout the nucleoplasm and cytoplasm in most neurons but in two discrete subpopulations is either excluded from the nucleus of neurons engaged in new viral DNA synthesis or forms discrete nuclear puncta in infected neurons where DNA synthesis is absent. It is tempting to imagine these puncta correspond to locations of viral genomes but this needs to be addressed experimentally. Nonetheless, differential localization of the host Gadd45b protein provides a useful marker to distinguish between successful and abortive reactivation events. The presence of multiple neuronal subtypes may be a confounding factor that contributes to the heterogeneity evident in both establishment of latency and subsequent reactivation. With the exception of the Gadd45b study, this has not been investigated extensively in vitro. Moreover, because these are low-frequency events, they are essentially invisible in bulk analyses and require simultaneous detection of viral and host markers at the level of individual neurons.

## 13. Closing Thoughts

The number of researchers using neuronal infection models to study α-HV latency and reactivation is at an all-time high and seems poised to continue increasing. This surge in popularity is propelled by the wealth of mechanistic insights that have emerged in the last few years and a growing appreciation for the potential of in vitro models to address long-standing mechanistic questions that are out of reach otherwise. Technological advances have played a major role in expanding the experimental possibilities. Innovations include the use of fluorescent reporter viruses that provide readouts in real time, a growing arsenal of chemical inhibitors with clearly defined molecular targets, the ease by which lentiviral transduction and RNA interference can be used to deplete host and viral gene products, and a wealth of exquisitely sensitive assay tools. It is encouraging to see many researchers embracing emerging technologies such as organoid culture, super-resolution imaging and single molecule detection. As often happens in science, technical innovations expand the sorts of biological questions that can be asked. Increasingly, studies are shifting from population-level (bulk) measurements to assays that trace processes or events in individual cells. In time, this precision will almost certainly extend to the level of individual viral genomes, something that would have been inconceivable not so long ago. Finally, there is growing societal and political pressure to ‘refine, reduce, and replace’ the use of animals in biomedical research and some investigators may be drawn to human iPSC and ESC-based models for this reason alone [116]. With so much to look forward to, our thoughts return to Randy Cohrs and the satisfaction he would have found in harnessing these exciting technologies to solve the puzzles in virology that interested him so deeply.

## Figures and Tables

**Figure 1 viruses-14-01209-f001:**
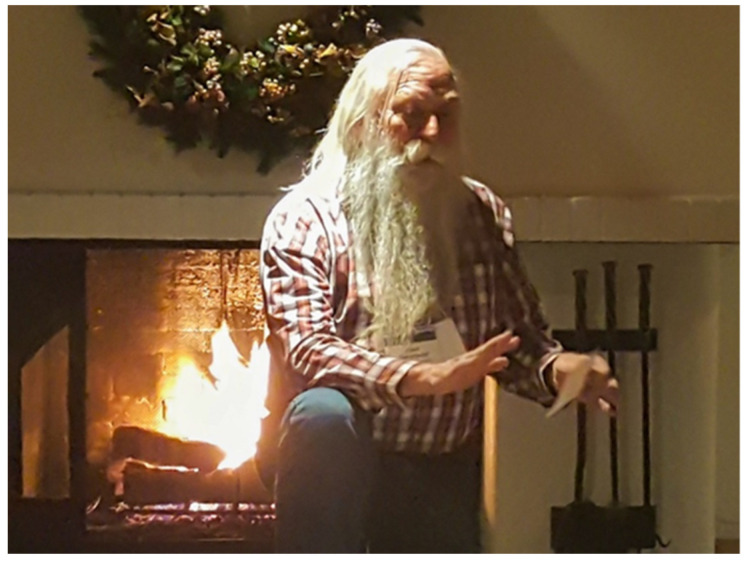
Randy Cohrs in his element reading out discussion topic suggestions in front of a roaring fire on the last night of CALS 2017. Photograph generously provided with permission by CALS.

**Figure 2 viruses-14-01209-f002:**
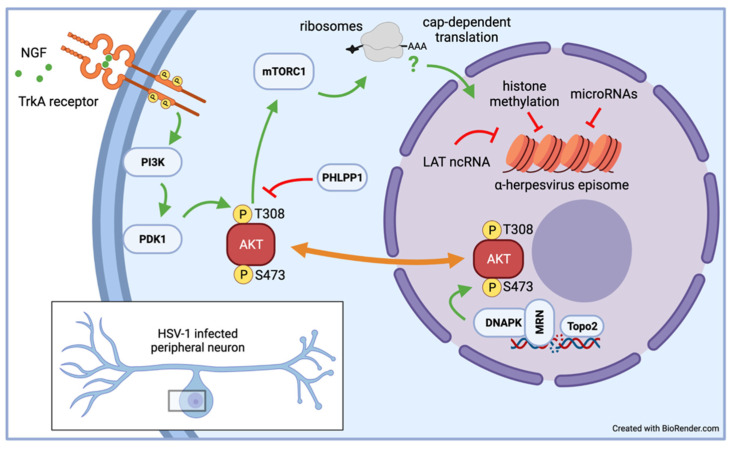
Host kinase AKT integrates NGF signaling with maintenance of chromosomal integrity to maintain HSV-1 latency. Studies in rat SCG neurons have shown that the AKT integrates external signaling from NGF interacting with its cognate receptor TrkA and intracellular DNA damage signaling initiating at topoisomerase 2 (Topo2)-induced DNA breaks. By shuttling between the cytoplasm and nucleus, AKT is kept in an active state through simultaneous phosphorylation of threonine-308 (T308^P^) by PDK1 in the cytoplasm and of serine 473 (S473^P^) by DNAPK in the nucleus. The AKT-mTORC1 axis ensures continuous cap-dependent protein synthesis, which is required to maintain the HSV-1 genome in a transcriptionally repressed state. Silencing of viral productive cycle genes involves the combined activities of repressive facultative heterochromatin chromatin, microRNAs, and potentially by lncRNAs such as the viral latency-associated transcript (LAT).

**Figure 3 viruses-14-01209-f003:**
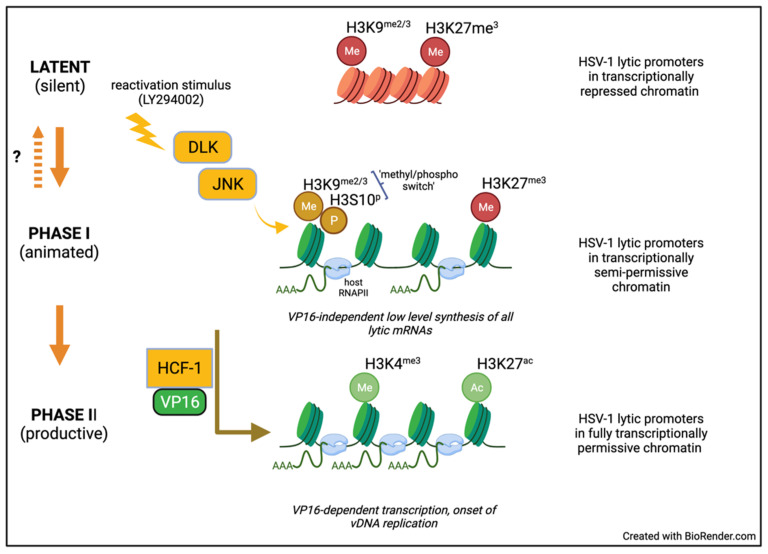
Biphasic model of HSV-1 reactivation. Studies using the rat and mouse SCG neuron infection models have found that HSV-1 reactivation proceeds through two mechanistically distinct steps referred to as Phase I and Phase II. Stresses such as interruption of growth factor signaling activates dual leucine zipper kinase (DLK) which in turn activate JNK resulting in phosphorylation of neuronal transcription factors and the repressive chromatin associated with the viral genome. JNK targets include serine-10 of histone H3 (H3S10^P^), which is adjacent to methylated lysine-9 (H3K9^me3^), a mark of repressive chromatin. This combinatorial mark (‘methyl/phospho switch’) renders chromatin permissive to RNAPII transcription, allowing widescale but low-level expression of viral productive cycle mRNAs. This transient animation of the viral transcriptome is termed Phase I. Although not yet test directly, dephosphorylation could potentially return animated genomes to their original transcriptionally silent state. The viral regulator VP16 is synthesized during Phase I but accumulates in the cytoplasm and does not influence viral gene expression. However, in a few neurons, VP16 is transported into the nucleus together with the coactivator HCF-1 and together recruit additional chromatin modifiers that remove repressive modifications and likely add activating marks such as methylation on lysine-4 of histone (H3K4^me3^) and/or acetylation of lysine 27 (H3K27^ac^) rendering the chromatin fully permissive for transcription. This VP16-dependent step is termed Phase II and results in higher levels of productive cycle gene transcription and permits the viral DNA genome amplification, which is not detected in Phase I.

**Figure 4 viruses-14-01209-f004:**
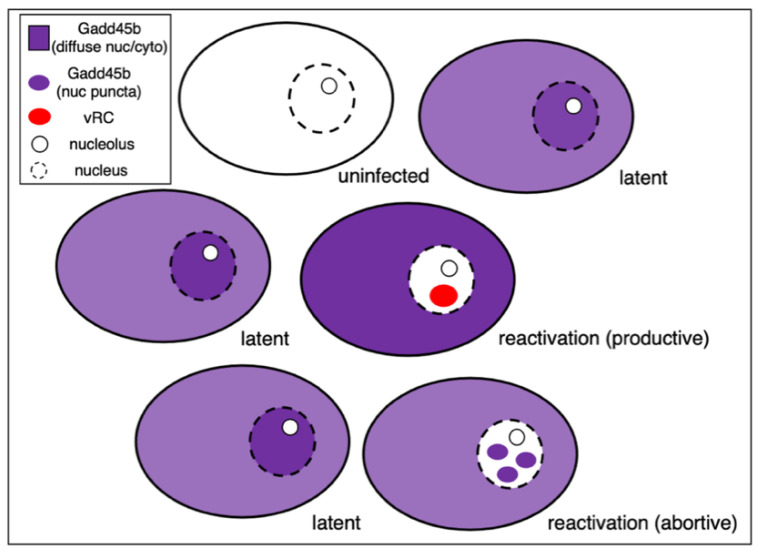
Subcellular localization of the Gadd45b protein acts as a marker of HSV-1 reactivation heterogeneity within a neuronal population. Schematic showing the different patterns of Gadd45b protein (purple) accumulation within the cell bodies of latently-infected rat SCG neurons treated with reactivation inducer LY294002. Protein levels are elevated only in latently infected neurons and is distributed throughout the cytoplasm and nucleoplasm. In a small number of neurons, Gadd45b is excluded from the nucleus and these neurons contain EdU-positive foci (red) corresponding to sites of active viral DNA replication (vRC) a marker of Phase II. These individual neurons are considered to be engaged in successful reactivation. In another rare population, Gadd45b accumulates as discrete puncta with the nucleus. These neurons are always EdU-negative and likely have not entered into Phase II and represent unsuccessful or abortive reactivation events. For full details see Hu et al., 2021 [45].

## Data Availability

Not applicable.

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
