# Peer review of "Impact of Cultured Neuron Models on α-Herpesvirus Latency Research"

_viruses, 2022, doi:10.3390/v14061209_

Round 1

Reviewer 1 Report

Very informative review. Would be nice to elaborate on human
iPSC and ESC-based models and organoid challenges are currently ongoing.

Author Response

Thank you. I do discuss iPSC and ESC models in a number of places and also cite most if not all of the relevant papers. When I was planning this review I expected this to be a larger component but discovered that most studies address productive (acute) infection, rather than true latency and reactivation. This is also true for organoid models. Without a doubt, this is an area awaiting development, which I emphasize, and I hope this summary of considerations based on other models will contribute.

Reviewer 2 Report

The Review written by Angus Wilson is comprehensive, scientifically sound and well written. I don't have any major concerns and consider that it will be a great contribution to the field of alphaherpesvirus latency.

I only have one minor comment: Figure 3 depicting the subcellular localization of Gadd45b is not providing much novelty that was not already covered in reference 45 of this review. I would suggest to replace Figure 3 with a picture of the CALS "fireside chats" as part of the nice tribute to Randy Cohrs.

Author Response

Thank you for the kind words. I have reached out to others and have managed to track down a nice photograph of Randy holding court in front of the fireplace (new Figure 1). This is a great accompaniment to the prologue.  With all respect, I would like to retain the schematic showing the different configurations of Gadd45b (now Figure 4). I am not sure how many people will actually read the original paper and want readers to retain a mental image of this important finding. I think there's room for these four compact figures and one table.